

# The complete chloroplast genome sequence of strawberry (*Fragaria × ananassa* Duch.) and comparison with related species of Rosaceae

Hui Cheng[1], Jinfeng Li[2], Hong Zhang[1], Binhua Cai[1], Zhihong Gao[1], Yushan Qiao[1] and Lin Mi[2]

[1] Laboratory of Fruit Tree Biotechnology, College of Horticulture, Nanjing Agricultural University, Nanjing, China
[2] Laboratory of Fruit Tree, Zhenjiang Institute of Agricultural Sciences in Hilly Area of Jiangsu Province, Jurong, China

Corresponding authors
Yushan Qiao,
qiaoyushan@njau.edu.cn
Lin Mi, jsrmn6217@sina.com

## ABSTRACT

Compared with other members of the family Rosaceae, the chloroplast genomes of *Fragaria* species exhibit low variation, and this situation has limited phylogenetic analyses; thus, complete chloroplast genome sequencing of *Fragaria* species is needed. In this study, we sequenced the complete chloroplast genome of *F. × ananassa* 'Benihoppe' using the Illumina HiSeq 2500-PE150 platform and then performed a combination of *de novo* assembly and reference-guided mapping of contigs to generate complete chloroplast genome sequences. The chloroplast genome exhibits a typical quadripartite structure with a pair of inverted repeats (IRs, 25,936 bp) separated by large (LSC, 85,531 bp) and small (SSC, 18,146 bp) single-copy (SC) regions. The length of the *F. × ananassa* 'Benihoppe' chloroplast genome is 155,549 bp, representing the smallest *Fragaria* chloroplast genome observed to date. The genome encodes 112 unique genes, comprising 78 protein-coding genes, 30 tRNA genes and four rRNA genes. Comparative analysis of the overall nucleotide sequence identity among ten complete chloroplast genomes confirmed that for both coding and non-coding regions in Rosaceae, SC regions exhibit higher sequence variation than IRs. The Ka/Ks ratio of most genes was less than 1, suggesting that most genes are under purifying selection. Moreover, the mVISTA results also showed a high degree of conservation in genome structure, gene order and gene content in *Fragaria*, particularly among three octoploid strawberries which were *F. × ananassa* 'Benihoppe', *F. chiloensis* (GP33) and *F. virginiana* (O477). However, when the sequences of the coding and non-coding regions of *F. × ananassa* 'Benihoppe' were compared in detail with those of *F. chiloensis* (GP33) and *F. virginiana* (O477), a number of SNPs and InDels were revealed by MEGA 7. Six non-coding regions (*trnK-matK*, *trnS-trnG*, *atpF-atpH*, *trnC-petN*, *trnT-psbD* and *trnP-psaJ*) with a percentage of variable sites greater than 1% and no less than five parsimony-informative sites were identified and may be useful for phylogenetic analysis of the genus *Fragaria*.

## INTRODUCTION

The chloroplast, which is considered to have originated from free-living cyanobacteria through endosymbiosis, plays an essential role in photosynthesis and many biosynthetic activities (*Keeling, 2004*). Most chloroplast genomes of angiosperms exhibit a highly conserved organization with a typical quadripartite structure that includes two copies of inverted repeats (IRs), separated by large (LSC) and small (SSC) single-copy (SC) regions (*Palmer, 1991*; *Jansen et al., 2005*). In general, the chloroplast genomes of angiosperms encode 110–130 genes with a size range of 120–160 kb (*Palmer, 1985*). Variation in genome size can be attributed to IR expansion/contraction or even loss (*Ma et al., 2014*; *Zhang et al., 2014*; *Lei et al., 2016*). Although chloroplast DNA (cpDNA) is inherited maternally in most angiosperms, cpDNA transmission in *Medicago sativa* is reported to be biparental or paternal (*Smith, Bingham & Fulton, 1986*; *Schumann & Hancock, 1989*), and paternal inheritance has been demonstrated in *Actinidia chinensis* (*Testolin & Cipriani, 1997*). Compared with the nuclear genome, the chloroplast genome is small, and the rate of nucleotide substitutions is so low that the chloroplast genome is considered to be an ideal system for studies on phylogeny (*Wei et al., 2005*). In addition, chloroplast transformation presents the advantages of producing high protein levels, site-specific integration of transgenes, and a lack of posttranscriptional gene silencing, making it an environmentally friendly strategy for plant genetic engineering (*Daniell, Khan & Allison, 2002*; *Bock, 2014*).

The family Rosaceae includes approximately 3,000 species from 90 genera distributed throughout the world, with particular enrichment in the North Temperate Zone (*Potter et al., 2007*), and many species of Rosaceae exhibit important economic value, such as common fruits, including apple (*Malus*), pear (*Pyrus*), peach (*Prunus*) and strawberry (*Fragaria*) as well as ornamentals, e.g., *Rosa* and *Spiraea*. The assembled nuclear genomes of *Malus × domestica* (*Velasco et al., 2010*), seven *Fragaria* species (*Shulaev et al., 2011*; *Hirakawa et al., 2014*; *Tennessen et al., 2013*), *Prunus mume* (*Zhang et al., 2012*), *Pyrus bretschneideri* (*Wu et al., 2013*), *Prunus persica* (*Verde et al., 2013*), and *Rubus occidentalis* (*VanBuren et al., 2016*) have been reported, providing valuable information for evolutionary classification. Nevertheless, due to apomixis, hybridization and assumed rapid radiation, the phylogenetic relationships among Rosaceae species have long been uncertain (*Potter et al., 2007*; *Campbell et al., 2007*; *Lo & Donoghue, 2012*). With the rapid development of next-generation sequencing, researchers recently sequenced 125 new transcriptomic and genomic datasets and identified hundreds of nuclear genes to reconstruct a well-resolved Rosaceae phylogeny (*Xiang et al., 2017*). Moreover, 130 complete chloroplast genomes in Rosaceae have also been sequenced, and the phylogenetic relationships among members of this family have been thoroughly analyzed (*Zhang et al., 2017*).

The genus *Fragaria* belongs to subtribe Fragariinae within tribe Potentilleae of subfamily Rosoideae (*Potter et al., 2007*; *Xiang et al., 2017*) and is comprised of one cultivated (*F. × ananassa*) and 24 wild species (*Staudt, 2009*; *Hummer, Nathewet & Yanagi, 2009*). *Fragaria* species exhibit natural variation in ploidy ranging from diploid to decaploid (*Hummer, Nathewet & Yanagi, 2009*; *Hummer, 2012*), although chloroplast DNA is unaffected by such changes in ploidy, which can complicate phylogenetic analyses (*Palmer, 1986*). Moreover, as haplotype analysis supports maternal inheritance of the

chloroplast genome in *Fragaria* (*Honjo et al., 2009*; *Davis et al., 2010*), phylogenetic analyses of *Fragaria* have been attempted using chloroplast genome sequences (*Harrison, Luby & Furnier, 1997*; *Potter, Luby & Harrison, 2000*; *Lin & Davis, 2000*; *Njuguna et al., 2013*; *Govindarajulu et al., 2015*). Although *Fragaria* exhibits limited variation in chloroplast sequences (*Njuguna, 2010*), comparative analyses of *Fragaria* using the entire chloroplast genome can provide comprehensive genetic information, for example, on InDels and nucleotide substitutions, which can be utilized as molecular markers and for diversity analyses (*Cho et al., 2015*). To date, seven complete chloroplast genomes of *Fragaria* have been released by National Centre for Biotechnology Information (NCBI, https://www.ncbi.nlm.nih.gov/), for one accession each of the octoploids *F. chiloensis* (GP33, PI612489) (GenBank: JN884816), and *F. virginiana* (O477, PI657873) (GenBank: JN884817); diploid *F. vesca* ssp. *vesca* (Hawaii 4, PI551572) (GenBank: JF345175); three accessions of diploid *F. vesca* ssp. *bracteata* (MRD30, PI 664465; MRD102; LNF40) (GenBank: KC507755, KC507756, and KC507757); diploid *F. pentaphylla* (GenBank: KY434061), and a partial 130 kb chloroplast genome assembly of *F. vesca* ssp. *americana* (cp130096) as submitted GenBank (GU363535) (*Davis et al., 2010*). Thus, enrichment of complete chloroplast genomes is necessary to study evolution in *Fragaria*.

Cultivated strawberry (*F. × ananassa* Duch.) is one of the most economically important fruit crops in the world. It originated from accidental hybridization between *F. virginiana* and *F. chiloensis* in Europe during the early to mid-1700s, and systematic breeding using a small number of native and cultivated clones began in England and North America in the 1800s (*Darrow, 1966*). Wild strawberries have recently been employed to increase genetic diversity (*Noguchi, 2011*), though most modern strawberry cultivars are the progeny of *F. × ananassa* germplasm (*Honjo et al., 2009*; *Hancock, 2008*; *Luby et al., 2008*). *F. × ananassa* 'Benihoppe' (Registration no. 10371 in Japan, http://www.hinsyu.maff.go.jp/) was selected from Akihime × Sachinoka progenies in Shizuoka Prefecture, Japan, in 1994. This cultivar exhibits the characteristics of large size, rich flavor, firm texture and high yield (*Takeuchi et al., 1999*) and has become one of the main strawberry cultivars grown in China. The research of transgenic 'Benihoppe' strawberry via *Agrobacterium*-mediated to nuclear genome has been reported (*Wang et al., 2014*; *Gu et al., 2015*; *Gu et al., 2017*). However, the lack of a complete chloroplast genome sequence is one of the major limitations restricting the development of chloroplast genetic engineering.

Here, we report the first complete chloroplast genome of cultivated strawberry (*F. × ananassa* 'Benihoppe') based on next-generation sequencing methods (Illumina HiSeq 2500-PE150). In addition to describing the characteristics of the chloroplast genome, we conducted comparative analysis against nine other Rosaceae species, including *Fragaria* species in particular. The generation of the complete chloroplast genome of *F. × ananassa* 'Benihoppe' is significant for phylogenetic and evolutionary research within *Fragaria* and provides valuable data for chloroplast genetic engineering and understanding molecular evolution.

## MATERIALS AND METHODS

### Plant material, DNA sequencing and genome assembly

Approximately 100 g of fresh young leaves of *F*. × *ananassa* 'Benihoppe' was collected from the Zhenjiang Institute of Agricultural Sciences in a Hilly Area of Jiangsu, Jurong, China. The voucher specimens were deposited in the laboratory of Fruit Tree Biotechnology of Nanjing Agricultural University. Chloroplast DNA was extracted using the high-salt saline plus Percoll gradient method of *Vieira et al. (2014)*. A paired-end library was constructed from 50 ng of purified cpDNA according to the manufacturer's instructions (Illumina, San Diego, CA, USA). The library, which contained an insert size of 350 bp, was sequenced using the Illumina HiSeq 2500-PE150 platform by Beijing Novogene Bioinformatics Technology Co., Ltd. (Beijing, China). MITObim v1.8 (*Hahn, Bachmann & Chevreux, 2013*) was utilized for *de novo* genome assembly, and the chloroplast genome reads were aligned to closely related cpDNA sequences from *F. vesca* ssp. *vesca* Hawaii 4 (JF345175). Different k-mer sizes were tested, among which 31 bp produced the best results and was used to generate the final assembly in terms of the single longest scaffold length. The junctions between SC and IR regions were verified through polymerase chain reaction (PCR) amplification using sequence-specific primers (File S1). The PCR products were sequenced via Sanger sequencing.

### Genome annotation and codon usage

The Dual Organellar GenoMe Annotator (DOGMA; http://dogma.ccbb.utexas.edu/, *Wyman, Jansen & Boore, 2004*) was employed to annotate the *F*. × *ananassa* 'Benihoppe' chloroplast genome. The initial annotations and putative start, stop, and intron positions were checked manually based on comparison with homologous genes in other *Fragaria* chloroplast genomes available in the GenBank database. Additionally, tRNA genes were identified using tRNAscan-SE 1.21 (http://lowelab.ucsc.edu/tRNAscan-SE/; *Schattner, Brooks & Lowe, 2005*) and ARAGORN (*Laslett & Canback, 2004*). A circular chloroplast genome map of *F*. × *ananassa* 'Benihoppe' was constructed using the online tool OGDRAW (http://ogdraw.mpimp-golm.mpg.de; *Lohse, Drechsel & Bock, 2007*). GC content, codon usage and relative synonymous codon usage (RSCU) were analyzed with MEGA 7 software (*Kumar et al., 2008*).

### Repeat structure and simple sequence repeats (SSRs)

The sizes and locations of forward, reverse, palindromic and complementary repeats were determined with the REPuter program (*Kurtz et al., 2001*). The minimum identity and size of the repeats were limited to 90% (Hamming distance of 3) and 30 bp, respectively. SSRs in the chloroplast genome were detected using MISA (*Thiel et al., 2003*) with the following parameters: minimum SSR motif length of 10 bp and repeat lengths of mono-10, di-5, tri-4, tetra-3, penta-3 and hexa-3.

### Comparison with other Rosaceae chloroplast genomes

One species was selected from each of the four most important fruit tree or ornamental species (*Malus Mill.*, *Pyrus L.*, *Prunus L.*, and *Rosa L.*) of Rosaceae and from the genus *Fragaria*, including *F. chiloensis* (GP33), *F. virginiana* (O477), *F. vesca* ssp. *vesca* (Hawaii 4),

*F. vesca* ssp. *bracteata* (MRD30) and *F. pentaphylla* (KY434061). The complete chloroplast genome of *F. × ananassa* 'Benihoppe' was employed as a reference and was compared with the chloroplast genomes of the nine other species using mVISTA software in the Shuffle-LAGAN mode (*Frazer et al., 2004*).

### Nucleotide substitution in coding regions

All 78 functional protein-coding genes were extracted from the six *Fragaria* species (Rosoideae), *Rosa roxburghii* (Rosoideae) (KX768420) and *Prunus persica* 'Nemared' (Amygdaloideae) (HQ336405), and 77 protein-coding genes from *Malus prunifolia* (MPRUN20160302) (Amygdaloideae) (KU851961) and *Pryus pyrifolia* 'Hosui' (Amygdaloideae) (AP012207) were employed because the *psbL* gene was not annotated. Each exon was aligned with those of *F. × ananassa* 'Benihoppe' using ClustalX v2.1 (*Thompson et al., 1997*). The alignment file was then analyzed with DnaSP v5 (*Librado & Rozas, 2009*) to calculate synonymous (Ks) and nonsynonymous (Ka) substitution rates.

### cpDNA marker identification in *Fragaria*

The seven complete chloroplast genomes of *Fragaria* species that have been released by NCBI (as above) as well as two nearly complete chloroplast genomes which were *F. mandshurica* (fc199s6) (KC507760) and *F. iinumae* (fc199s5) (KC507759) and the chloroplast genome of cultivar 'Benihoppe' were used to identify rapidly evolving molecular markers that may be employed for phylogenetic analysis of *Fragaria*. As the coding regions are highly conserved, only fragments from non-coding regions were considered. Homologous regions were aligned using MEGA 7 and adjusted manually where necessary. Then, the percentage of variable sites for each region was calculated. The proportion of mutation events = (NS/L) × 100, where NS = number of nucleotide substitutions, and L = aligned sequence length (*Li et al., 2013*). Because parsimony-informative sites (PIS) are commonly used in phylogenetic analyses, the number of PIS was calculated as well.

To examine the phylogenetic applications of rapidly evolving molecular markers, the maximum parsimony (MP) method was employed to construct phylogenetic trees using MEGA 7 with the following parameters: gaps in the alignment treated as missing, 1,000 replicates for bootstrap support, and tree bisection-reconnection (TBR) branch swapping.

## RESULTS AND DISCUSSION

### Chloroplast genome assembly, organization, and gene content

In total, 276 Mb of 150-bp raw paired-end reads was retrieved and trimmed, and 241 Mb of high-quality short reads was finally employed to assemble the chloroplast genome, using a combination of the MITObim v1.8 *de novo* assembly and reference-guided (GenBank: JF345175) mapping of contigs to generate complete chloroplast genome sequences. Finally, the generated data were assembled into the single longest scaffold spanning the *F. × ananassa* 'Benihoppe' chloroplast genome. To validate the assembly, four junctions between SC and IR regions were confirmed through PCR amplification and Sanger sequencing. No mismatches or InDels were observed between the Sanger sequencing and the assembled genome, which verified the correctness of our genome sequencing and assembly results.

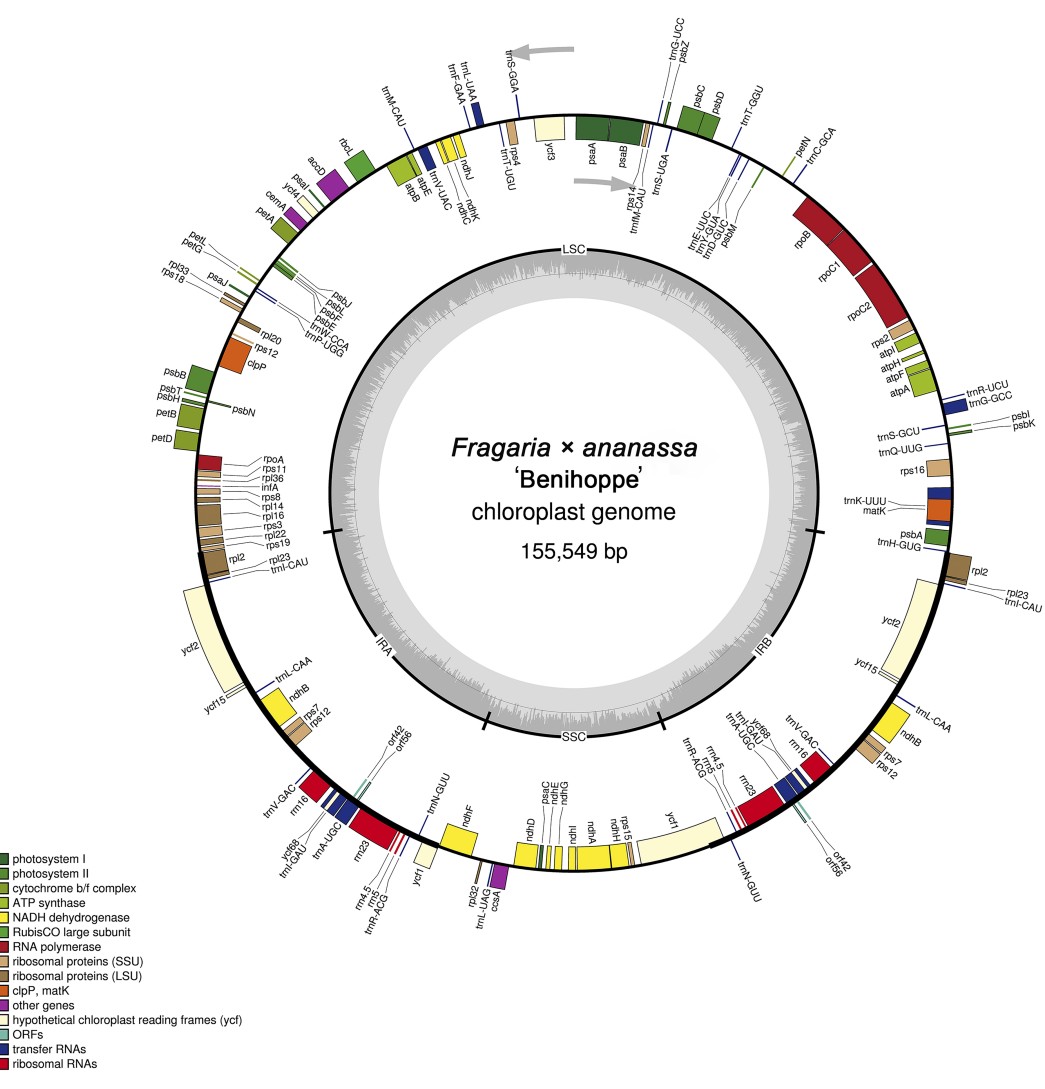

**Figure 1** **Gene map of the *F. × ananassa* 'Benihoppe' chloroplast genome.** Genes inside the circle are transcribed in the clockwise direction, and those outside are transcribed in the counter-clockwise direction. Color coding indicates genes of different functional groups. The dark-gray inner circle denotes the GC content, and the lighter-gray circle denotes the AT content.

The *F. × ananassa* 'Benihoppe' chloroplast genome is a typical circular double-stranded DNA molecule with a quadripartite structure; it is 155,549 bp in size and consists of IR (25,936 bp) regions separated by LSC (85,531 bp) and SSC (18,146 bp) regions (Fig. 1, Table 1). The GC content of the chloroplast genome is 37.23% (Table 1), and the GC contents of the LSC and SSC regions (35.12% and 31.14%) are lower than those of the IR regions (42.85%). The high GC contents in the IR regions are mainly due to the high GC contents of the four ribosomal RNA (rRNA) genes (55.43%) which is similar to most of other plants cp genomes (*Wang, Shi & Gao, 2013*; *Shen et al., 2016*; *Kong & Yang, 2017*).

Cheng et al. (2017), *PeerJ*, DOI 10.7717/peerj.3919

Peerj

**Table 1  Summary of the complete chloroplast genome characteristics of ten species in Rosaceae.**

| Species | Genome size (bp) | LSC size (bp) | SSC size (bp) | IR size (bp) | Number of genes | Protein-coding genes | tRNA genes | rRNA genes | Number of genes duplicated in IR | GC content (%) | GenBank no. | Reference |
|---|---|---|---|---|---|---|---|---|---|---|---|---|
| *F. × ananassa* 'Benihoppe' | 155,549 | 85,531 | 18,146 | 25,936 | 112 | 78 (7) | 30 (7) | 4 (4) | 18 | 37.23% | KY358226 | This article |
| *F. chiloensis* (GP33) | 155,603 | 85,566 | 18,147 | 25,945 | 112 | 78 (7) | 30 (7) | 4 (4) | 18 | 37.22% | JN884816 | *Salamone et al. (2013)* |
| *F. virginiana* (O477) | 155,621 | 85,585 | 18,146 | 25,945 | 112 | 78 (7) | 30 (7) | 4 (4) | 18 | 37.23% | JN884817 | *Salamone et al. (2013)* |
| *F. vesca* ssp. *vesca* (Hawaii 4) | 155,691 | 85,605 | 18,174 | 25,956 | 112 | 78 (7) | 30 (7) | 4 (4) | 18 | 37.21% | JF345175 | *Shulaev et al. (2011)* |
| *F. vesca* ssp. *bracteata* (MRD30) | 155,619 | 85,566 | 18,151 | 25,951 | 112 | 78 (7) | 30 (7) | 4 (4) | 18 | 37.23% | KC507755 | Unpublished |
| *F. pentaphylla* | 155,640 | 85,571 | 18,145 | 25,962 | 112 | 78 (7) | 30 (7) | 4 (4) | 18 | 37.25% | KY434061 | *Bai et al. (2017)* |
| *R. roxburghii* | 156,749 | 85,851 | 18,792 | 26,053 | 114 | 79 (7) | 31 (7) | 4 (4) | 18 | 37.23% | KX768420 | Unpublished |
| *M. prunifolia* (MPRUN20160302) | 160,041 | 88,119 | 19,204 | 26,359 | 111 | 77 (7) | 30 (7) | 4 (4) | 18 | 36.56% | KU851961 | *Bao et al. (2016)* |
| *P. pyrifolia* 'Hosui' | 159,922 | 87,901 | 19,237 | 26,392 | 111 | 77 (6) | 30 (7) | 4 (4) | 17 | 36.58% | AP012207 | *Terakami et al. (2012)* |
| *P. persica* 'Nemared' | 157,790 | 85,968 | 19,060 | 26,381 | 112 | 78 (5) | 30 (7) | 4 (4) | 16 | 36.76% | HQ336405 | *Jansen et al. (2011)* |

**Notes.**

LSC, large single copy; SSC, small single copy; IR, inverted repeat (A/B); bp, base pairs.
Figures in brackets denote the number of genes duplicated in IR.

There are 130 genes in the chloroplast genome of *F. × ananassa* 'Benihoppe', 112 of which are unique, including 78 protein-coding genes, 30 tRNA genes and 4 rRNA genes. The genes that are repeated in IRs comprise seven protein-coding genes, seven tRNA genes, and four rRNA genes (Fig. 1, Table 2). Among these genes, a single intron was detected in 15 genes (nine protein-coding genes and 6 tRNA genes), while two genes (*ycf3* and *clpP*) were found exhibit two introns each (File S2). The *trnK-UUU* gene harbors the largest intron (2,497 bp), which contains the *matK* gene, whereas the intron of *trnL-UAA* is smallest (422 bp). Two genes with internal stop codons (*ycf15* and *ycf68*) and one without a stop codon (*infA*) were annotated as pseudogenes. Absence or pseudogenization of these three genes has also been reported in other Rosaceae species, such as *Eriobotrya japonica* (*Shen et al., 2016*) and *Prinsepia utilis* (*Wang, Shi & Gao, 2013*). The *rps12* gene is a trans-spliced gene with a 5′ exon located in an LSC region and two 3′ exons located in IR regions. The complete chloroplast genome with gene annotations has been deposited in the NCBI GenBank database (accession number: KY358226).

Overall, 22,709 codons encoding 78 protein-coding genes were identified in the complete chloroplast genome and classified according to codon usage (File S3), among which 2,405 (10.59%) encode leucine (the most abundant amino acid), and 252 (1.11%) encode cysteine (the least abundant amino acid). RSCU analysis showed A/T contents of 53.89%, 61.84% and 70.17% at the first, second and third codon positions, respectively. This pattern of higher A/T bias at the third codon position is common in the chloroplast genomes of land plants (*Morton, 1998*; *Gurusamy & Seonjoo, 2016*).

## Repeat structure and SSR loci

A total of 39 repeat structures with a minimal length of 30 bp and minimal identity of 90% were found (File S4), including 14, 6, 2 and 17 forward, reverse, complementary, and palindromic structures, respectively. Among these structures, the longest is 67 bp and is located between *trnM-CAU* and *atpE*. Most of the repeat structures are located in intergenic regions (65.4%), while fewer than half are located in coding genes (21.8%; *ndhA*, *ycf2*, *psaA*, *psaB*, *trnS-GGA*, *psbJ*, *trnG-UCC* and *trnG-GCC*) or introns (12.8%; *ycf3*, *ndhB* and *clpP*).

Simple sequence repeats (SSRs) in chloroplast genomes have become valuable molecular markers because of their high degree of variation within an individual species, which is useful for linkage map construction and plant breeding (*Powell et al., 1995*; *Xue, Wang & Zhou, 2012*). In the *F. × ananassa* 'Benihoppe' chloroplast genome, 61 SSR loci with a length of at least 10 bp were detected, among which 38 (62.3%) are mononucleotide repeats; 16 (26.2%) are di-repeats; three (4.9%) are tri-repeats; and four are (6.6%) tetra-repeats. No pentanucleotides or hexanucleotides were found. Most of the observed mononucleotide repeat sequences consist of A/T motifs, whereas only one is composed of a G/C motif. Similarly, 93.75% of the dinucleotide repeat sequences consist of AT/TA motifs. The results showed that the SSRs exhibit a strong AT bias, which is consistent with other studies (*Lei et al., 2016*; *Kuang et al., 2011*). Among the 61 SSR loci, 44 are located in intergenic regions, eight in introns, and nine in coding regions of genes (Table 3).
**Table 2  List of annotated genes in the *F. × ananassa* 'Benihoppe' chloroplast genome.**

| Category | Gene group | Gene name | | | | |
|---|---|---|---|---|---|---|
| **Photosynthesis** | Subunits of photosystem I | *psaA* | *psaB* | *psaC* | *psaI* | *psaJ* |
| | Subunits of photosystem II | *psbA* | *psbB* | *psbC* | *psbD* | *psbE* |
| | | *psbF* | *psbH* | *psbI* | *psbJ* | *psbK* |
| | | *psbL* | *psbM* | *psbN* | *psbT* | *psbZ* |
| | Subunits of NADH dehydrogenase | *ndhA*[b] | *ndhB*[b,c] | *ndhC* | *ndhD* | *ndhE* |
| | | *ndhF* | *ndhG* | *ndhH* | *ndhI* | *ndhJ* |
| | | *ndhK* | | | | |
| | Subunits of cytochrome b/f complex | *petA* | *petB*[b] | *petD*[b] | *petG* | *petL* |
| | | *petN* | | | | |
| | Subunits of ATP synthase | *atpA* | *atpB* | *atpE* | *atpF* | *atpH* |
| | | *atpI* | | | | |
| | Large subunit of rubisco | *rbcL* | | | | |
| **Self-replication** | Proteins of large ribosomal subunit | *rpl2*[b,c] | *rpl14* | *rpl16*[b] | *rpl20* | *rpl23*[c] |
| | | *rpl32* | *rpl33* | *rpl36* | | |
| | Proteins of small ribosomal subunit | *rps2* | *rps3* | *rps4* | *rps7*[c] | *rps8* |
| | | *rps11* | *rps12*[b,c] | *rps14* | *rps15* | *rps16*[b] |
| | | *rps18* | *rps19* | | | |
| | Subunits of RNA polymerase | *rpoA* | *rpoB* | *rpoC1*[b] | *rpoC2* | |
| | Ribosomal RNAs | *rrn16*[c] | *rrn23*[c] | *rrn4.5*[c] | *rrn5*[c] | |
| | Transfer RNAs | *trnA-UGC*[b,c] | *trnC-GCA* | *trnD-GUC* | *trnE-UUC* | *trnF-GAA* |
| | | *trnG -GCC*[b] | *trnG-UCC* | *trnH-GUG* | *trnI-CAU*[c] | *trnI-GAU*[b,c] |
| | | *trnK-UUU*[b] | *trnL-CAA*[c] | *trnL-UAA*[b] | *trnL-UAG* | *trnfM-CAU* |
| | | *trnM-CAU* | *trnN-GUU*[c] | *trnP-UGG* | *trnQ-UUG* | *trnR-ACG*[c] |
| | | *trnR-UCU* | *trnS-GCU* | *trnS-GGA* | *trnS-UGA* | *trnT-GGU* |
| | | *trnT-UGU* | *trnV-GAC*[c] | *trnV-UAC*[b] | *trnW-CCA* | *trnY-GUA* |
| **Other genes** | Maturase | *matK* | | | | |
| | Protease | *clpP*[a] | | | | |
| | Envelope membrane protein | *cemA* | | | | |
| | Acetyl-CoA carboxylase | *accD* | | | | |
| | c-type cytochrome synthesis gene | *ccsA* | | | | |
| | Translation initiation factor | *infA*[d] | | | | |
| **Genes of unknown function** | Conserved hypothetical chloroplast ORF | *ycf1*[c] | *ycf2*[c] | *ycf3*[a] | *ycf4* | *ycf15*[c,d] |
| | | *ycf68*[c,d] | *orf42*[c] | *orf56*[c] | | |

**Notes.**
[a] Gene with two introns.
[b] Gene with one intron.
[c] Genes located in the inverted repeats.
[d] Pseudogene.

## Comparison with other chloroplast genomes from Rosaceae

Nine chloroplast genomes representing five genera in Rosaceae were compared with that of *F. × ananassa* 'Benihoppe' (Table 1). The length of the *Fragaria* chloroplast genomes ranges from 155,549 to 155,691 bp, with *F. vesca* ssp. *vesca* (Hawaii 4) exhibiting the largest chloroplast genome and *F. × ananassa* 'Benihoppe' the smallest. The length of the LSC

**Table 3  Distribution of simple sequence repeat (SSR) loci in the _F._ × _ananassa_ 'Benihoppe' chloroplast genome.**

| Repeat motif | Length (bp) | Number of SSRs | Start position[a,b] |
|---|---|---|---|
| A | 10 | 11 | 3,744*; 7,019; 7,609; 8,256; 26,933; 47,455; 60,427; 65,327 (_psbF_); 66,476; 69,482; 109,237; |
|  | 11 | 2 | 15,732; 139,818 |
|  | 12 | 2 | 7,853; 136,910* |
|  | 15 | 1 | 8,608 |
|  | 16 | 1 | 36,532 |
|  | 17 | 1 | 7,969 |
| T | 10 | 8 | 15,712; 25,631 (_rpoB_); 46,143; 55,594 (_atpB_); 61,734; 121,755*; 128,786 (_ycf1_); 131,836 |
|  | 11 | 6 | 12,219; 17,914 (_rpoC2_); 45,448; 60,613; 101,254; 119,816 |
|  | 12 | 2 | 27,869; 104,161* |
|  | 14 | 1 | 70,953 |
|  | 15 | 1 | 71,654 |
|  | 16 | 1 | 64,340 |
| G | 12 | 1 | 64,213 |
| AT | 10 | 5 | 7,065; 29,392; 37,199; 60,337; 120,666 |
| TA | 10 | 5 | 4,891; 6,971; 19,292 (_rpoC2_); 52,497; 121,687* |
|  | 12 | 5 | 1,663; 6,993; 7,053; 36,475; 60,325; |
| TC | 10 | 1 | 62,100 (_cemA_) |
| AAT | 12 | 1 | 127,596 (_ycf1_) |
| ATA | 12 | 1 | 154,754* |
| TAT | 12 | 1 | 86,317* |
| AAAT | 12 | 1 | 55,693 |
| AATA | 12 | 1 | 6,423 |
| ATGT | 12 | 1 | 79,222 (_rpoA_) |
| TATT | 12 | 1 | 72,668* |

Notes.
[a]The SSR-containing coding regions are indicated in parentheses.
[b]Asterisk denote the SSR-containing introns.

regions shows greater variation, ranging from 85,531 to 85,605 bp, with _F. vesca_ ssp. _vesca_ (Hawaii 4) exhibiting the longest, followed by _F. virginiana_ (O477), while _F._ × _ananassa_ 'Benihoppe' harbors the shortest (Table 1). However, the IR regions of diploid strawberries are longer than those of three octoploid strawberries. _F. pentaphylla_ (KY434061) exhibits the shortest SSC regions. Furthermore, the size of the _F._ × _ananassa_ 'Benihoppe' chloroplast genome is smaller than those of the other four species in Rosaceae, being approximately 4.5 kb, 4.4 kb, 2.2 kb and 1.2 kb smaller than those of _M. prunifolia_ (MPRUN20160302), _P. pyrifolia_ 'Hosui', _P. persica_ 'Nemared', and _R. roxburghii_ (KX768420), respectively. The differences in genome size can largely be attributed to variation in the length of SSC and IR regions (Table 1).

The results also revealed that the gene content and gene order of _F._ × _ananassa_ 'Benihoppe' are identical to those of the five previously reported the genus _Fragaria_ chloroplast genomes. Interestingly, the loss of a group II intron of the _atpF_ gene, as observed in _Fragaria_ (Table 2), has previously been reported for Malpighiales (_Daniell et al., 2008_) and _R. roxburghii_ (KX768420). However, the numbers of unique genes

found in the *F.* × *ananassa* 'Benihoppe', *R. roxburghii* (KX768420), *M. prunifolia* (MPRUN20160302), *P. pyrifolia* 'Hosui', and *P. persica* 'Nemared' chloroplast genomes were 112, 114, 111, 111 and 112, respectively, due to the absence of the *psbL* gene in *M. prunifolia* (MPRUN20160302) and *P. pyrifolia* 'Hosui', the absence of the *trnG-GCC* gene in *R. roxburghii* (KX768420), and the presence of three genes, *infA*, *trnP-GGG* and *trnM-CAU*, only in *R. roxburghii* (KX768420). The GC content among the ten species was similar, ranging from 36.56 to 37.25%, with the seven Rosoideae species all exhibiting a high GC content, of approximately 37.2% (Table 1).

The mVISTA program was employed to analyze the overall sequence identity among all ten Rosaceae members at the chloroplast genome level, using the annotation for *F.* × *ananassa* 'Benihoppe' as a reference (Fig. 2). The results showed high similarity among the *Fragaria* chloroplast genome sequences, particularly for *F.* × *ananassa* 'Benihoppe', *F. chiloensis* (GP33) and *F. virginiana* (O477). Among the other Rosaceae species, the *F.* × *ananassa* 'Benihoppe' chloroplast genome was most similar to that of *R. roxburghii* (KX768420) and most divergent from that of *P. persica* 'Nemared'. Overall, the results revealed SC regions to be more divergent than IR regions, with higher divergence being observed in non-coding regions than in coding regions, which is a common phenomenon in the chloroplast genomes of angiosperms (*Yao et al., 2015*; *Ni et al., 2016*; *Asaf et al., 2016*). The coding regions with marked differences include the *ycf1*, *matK* and *psaI* genes. The highest divergence in non-coding regions was found for *rps16-trnQ*, *petN-psbM*, *ndhC-trnV*, *petA-psbL* and *rpl32-ccsA*. These results are similar to those of other analyses performed in Rosaceae (*Wang, Shi & Gao, 2013*; *Shen et al., 2016*), suggesting that these regions evolve rapidly in Rosaceae.

## IR contraction and expansion

In general, IR regions are considered to be the most conserved regions in the chloroplast genome. Nevertheless, expansion and contraction of the border region between SC and IR regions are common during evolution and contribute to variation in chloroplast genome length (*Wang et al., 2008*; *Li et al., 2013*). Thus, the positions of LSC/IRA/SSC/IRB borders and the adjacent genes in the ten Rosaceae chloroplast genomes were aligned (Fig. 3). The SSC/IRB boundary of *F.* × *ananassa* 'Benihoppe' is consistent with those of the other *Fragaria* species. All of the genomes except for those of *F. vesca* ssp. *vesca* (Hawaii 4) and *F. pentaphylla* (KY434061) exhibit IRA/SSC boundaries of the same length, and due to contraction of the IR region at the IRB/LSC boundary, *F.* × *ananassa* 'Benihoppe' exhibits the shortest IR region among the six *Fragaria* species.

Compared with those of other Rosaceae species, the *rps19* genes of *Fragaria* species and *R. roxburghii* (KX768420) are shifted to an LSC region with a 12–21 bp gap. However, the *rps19* genes of *M. prunifolia* (MPRUN20160302), *P. pyrifolia* 'Hosui' and *P. persica* 'Nemared' extend from the LSC to the IRA region, showing variability of 120–182 bp, resulting in the presence of an *rps19* pseudogene of the same length in IRB. The SSC/IRB boundary extends to the *ycf1* coding region, ranging from 1,051 bp (*P. persica* 'Nemared') to 1,106 bp (*R. roxburghii*, KX768420), leading to a nonfunctional *ycf1* gene in IRA. The *ndhF* gene is located entirely in the SSC region in Rosoideae species but varies in distance from

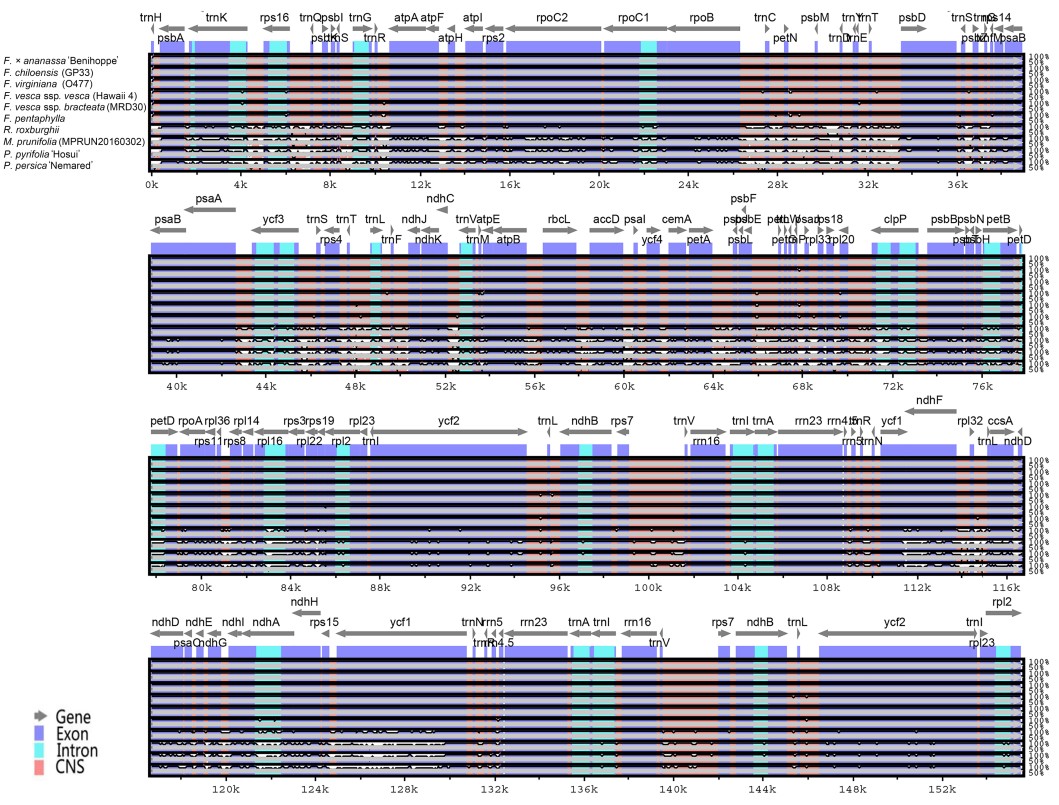

**Figure 2 Whole-chloroplast-genome alignments for nine Rosaceae species obtained using the mVISTA program, with the _F._ × _ananassa_ 'Benihoppe' chloroplast genome as the reference.** The Y-scale indicates identity from 50% to 100%. Gray arrows indicate the position and direction of each gene. Red indicates non-coding sequences (CNS); blue indicates the exons of protein-coding genes (exon); and lime green indicates the introns of protein-coding genes (intron).

the IRA/SSC border. However, in _M. prunifolia_ (MPRUN20160302), _P. pyrifolia_ 'Hosui', and _P. persica_ 'Nemared', part of the _ndhF_ gene is located in IRA. In general, the position of the _trnH_ gene in the chloroplast genome is quite conserved between monocot and dicot species. In monocots, the _trnH_ gene is located in the IR region, whereas it is located in the LSC region in dicots (_Asano et al., 2004_). In all of the analyzed genomes, the _trnH_ gene is located in the LSC region, although its distance from the IRB/LSC junction ranges from 0 to 101 bp. Overall, a similar pattern of expansion and contraction of IR/SC regions was observed among the _Fragaria_ species and _R. roxburghii_ (KX768420), differing from _M. prunifolia_ (MPRUN20160302), _P. pyrifolia_ 'Hosui' and _P. persica_ 'Nemared' (Fig. 3).

## Selection pressure on the _F._ × _ananassa_ 'Benihoppe' chloroplast genome

The Ka/Ks ratio was calculated for 78 protein-coding genes in all nine chloroplast genomes, with a value of 0 indicating neutral selection. The Ka/Ks ratio of the _Fragaria_ chloroplast genomes was typically calculated to be 0, except for six genes in _F. vesca_ ssp. _vesca_ (Hawaii 4) (_rpoC2_, _ndhD_, _ndhF_, _psbB_, _ycf1_ and _ycf4_), three genes in _F. vesca_ ssp. _bracteata_ (MRD30)

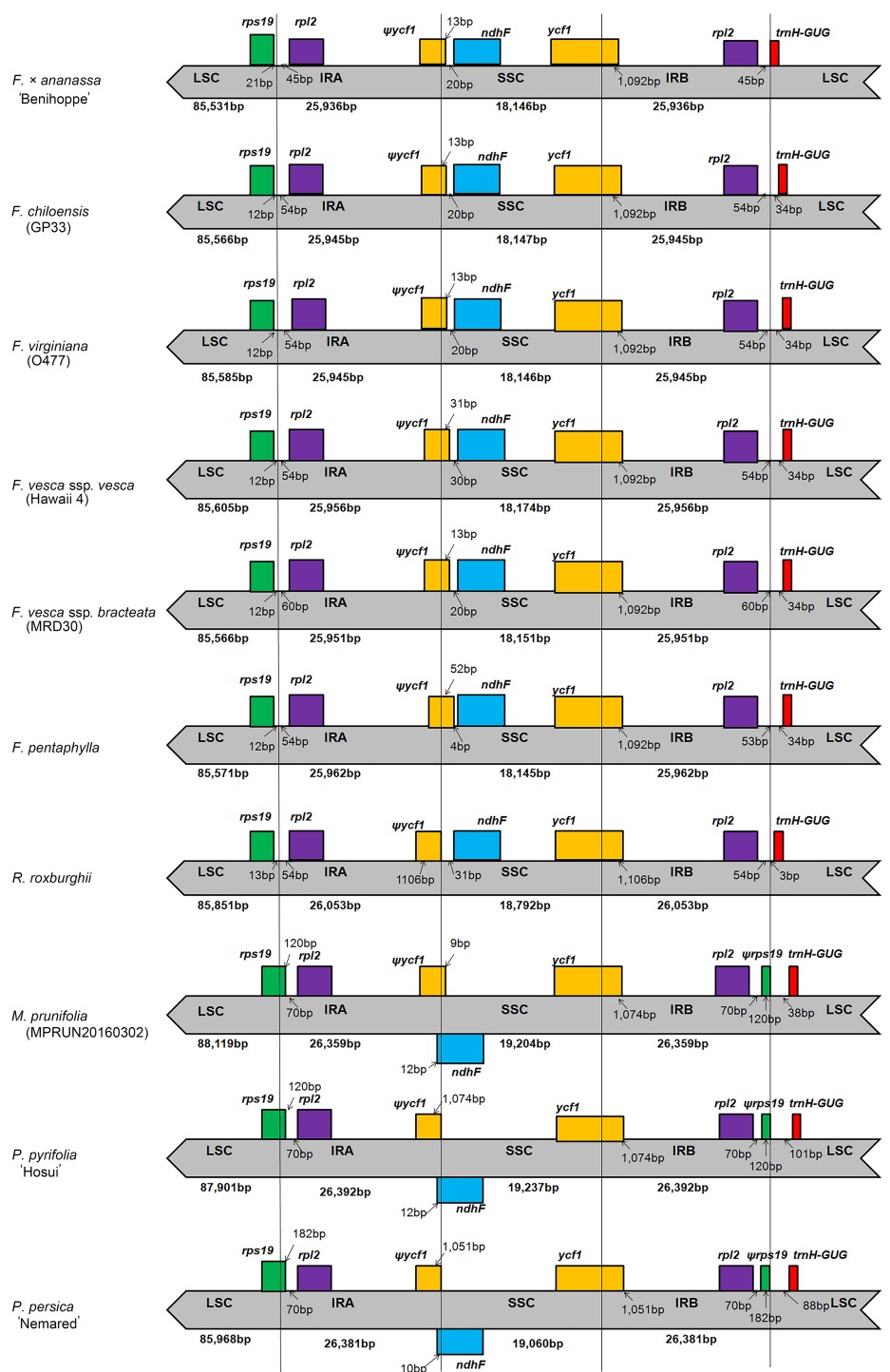

**Figure 3  Comparison of the borders of LSC, SSC and IR regions in ten Rosaceae chloroplast genomes.**

Table 4   Ka/Ks ratio of protein-coding genes from four Rosaceae species for comparsion with *Fragaria*.

| Region | *Fragaria* vs *Rosa* | *Fragaria* vs *Malus* | *Fragaria* vs *Pyrus* | *Fragaria* vs *Prunus* |
|--------|----------------------|------------------------|------------------------|-------------------------|
| LSC | 0.14101 | 0.11975 | 0.11924 | 0.11595 |
| IR | 0.11607 | 0.11744 | 0.12212 | 0.13104 |
| SSC | 0.14942 | 0.15972 | 0.15895 | 0.14729 |

(*ndhF*, *ycf1*, and *ycf4*) and thirteen genes in *F. pentaphylla* (KY434061) (*rpoC1*, *rpoC2*, *atpB*, *atpH*, *ndhA*, *ndhD*, *ndhF*, *ndhH*, *petA*, *psbB*, *rbcL*, *ycf1* and *ycf4*) (File S5).

   Among the protein-coding genes in the chloroplast genomes of the other Rosaceae species, the Ka/Ks ratio was observed to be highest in genes within the SSC regions (Table 4). In the comparison of *Fragaria* with *Rosa* and *Malus*, the lowest Ka/Ks ratio was found in the IR region. However, in the comparison of *Pyrus* and *Prunus*, the LSC region showed the lowest Ka/Ks ratio (Fig. 4, Table 4). The lowest Ka/Ks ratio was observed for genes encoding subunits of ATP synthase, subunits of the cytochrome b/f complex, subunits of photosystem II and the large subunit of RuBisCO (File S5). With the exception of the *rpl16* gene of *Rosa*, the Ka/Ks ratio of all genes was found to be less than 1, suggesting purifying selection on these genes (Fig. 4).

## Variation in chloroplast DNA in three octoploid strawberries

The complete chloroplast genomes were found to be most similar among the three octoploid strawberries. However, when the sequences of the coding and non-coding regions of *F. × ananassa* 'Benihoppe' were compared in detail with those of *F. chiloensis* (GP33) and *F. virginiana* (O477), a number of SNPs and InDels were revealed (Table 5).

   In total, 35 SNPs (26 transversions and 9 transitions) were identified between the complete *F. × ananassa* 'Benihoppe' and *F. chiloensis* (GP33) chloroplast genomes, which were found in all types of regions (23 in LSC, 9 in SSC and 3 in IR regions). Two SNPs in the *rpoB* and *ndhF* genes represent synonymous changes, whereas the other six SNPs in four other genes (*accD*, *ndhH*, *ycf1* and *ycf4*) are nonsynonymous and may alter the encoded protein's primary structure. Overall, 18 InDels between 1 and 19 bp in length were found, including 15 within LSC regions. In contrast, 23 SNPs (17 transversions and 6 transitions) were identified between the *F. × ananassa* 'Benihoppe' and *F. virginiana* (O477) chloroplast genomes, among which 21 are located in LSC and 2 in SSC regions, while none were found in IR regions. Three nonsynonymous SNPs were found in the *petB*, *ycf1* and *ycf4* genes, while three synonymous SNPs were found in *rps8*, *rpoB*, and *psbA*. Eight InDels were observed (5 insertions and 3 deletions), all but one of which is located within the LSC region.

   Regardless of location or subspecies, all *F. virginiana* individuals share the same SNPs in two genes, *petD* (G) and *ndhF* (A), differing from *F. chiloensis* at these positions (*petD*: A; *ndhF*: (G) (*Salamone et al., 2013*). Furthermore, *Honjo et al. (2009)* examined two non-coding regions and concluded that cultivar 'Benihoppe' exhibits haplotype V, which is consistent with its female parent Akihime. Based on our SNP analysis, two genes (*petD* and *ndhF*) and two non-coding regions (*trnL-trnF* and *trnR-rrn5*) are consistent with

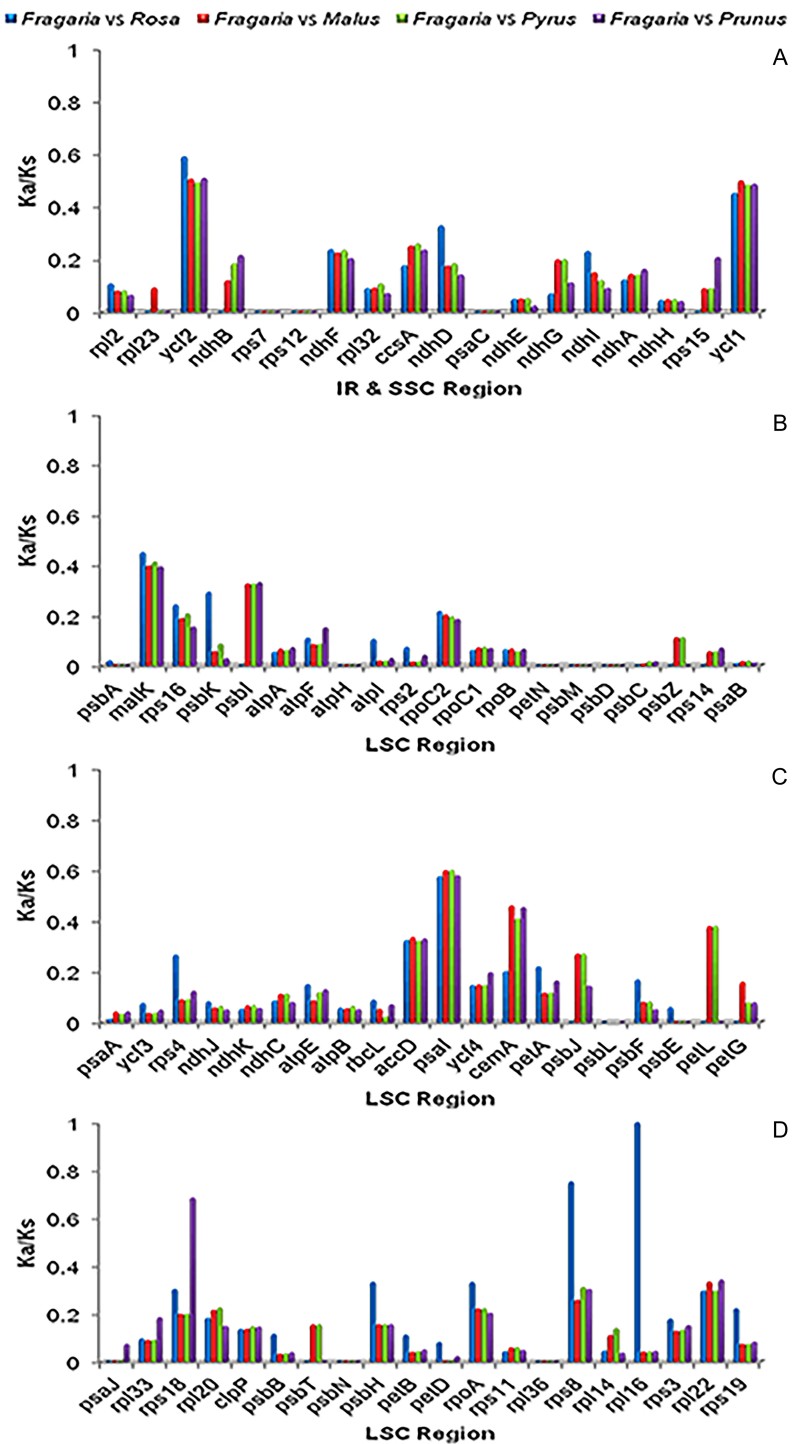

**Figure 4** **Ka/Ks ratios of 78 protein-coding genes in *Fragaria, Rosa, Malus, Pyrus* and *Prunus*.** Blue boxes indicate the Ka/Ks ratio for *Fragaria* vs. *Rosa*; red, *Fragaria* vs. *Malus*; green, *Fragaria* vs. *Pyrus*; and purple, *Fragaria* vs. *Prunus*.

**Table 5  SNPs and InDels among the *F.* × *ananassa* 'Benihoppe', *F. chiloensis* (GP33) and *F. virginiana* (O477) chloroplast genomes.**

| Number | Type | Position | Location | Nucleotide position[b] | *F.* × *ananassa* 'Benihoppe' | *F. chiloensis* (GP33) | *F. virginiana* (O477) |
|---|---|---|---|---|---|---|---|
| 1 | SNP | LSC/*trnK*-*rps16* | CNS[a] | 4,274 | C | A | C |
| 2 | SNP | LSC/*rps16*-intron | CNS | 5,974 | A | C | A |
| 3 | SNP | LSC/*rps16*-*trnQ* | CNS | 6,982 | T | A | T |
| 4 | SNP | LSC/*trnQ*-*psbK* | CNS | 7,609 | A | C | A |
| 5 | InDel | LSC/*trnS*-*trnG* | CNS | 8,635–8,636 | – | A | A |
| 6 | SNP | LSC/*trnG*-intron | CNS | 9,309 | G | T | G |
| 7 | SNP | LSC/*trnG*-*trnR* | CNS | 9,834 | T | T | A |
| 8 | InDel | LSC/*rps2*-*rpoC2* | CNS | 15,742–15,743 | – | A | – |
| 9 | SNP | LSC/*rpoB*-*trnC* | CNS | 26,855 | C | C | A |
| 10 | InDel | LSC/ *rpoB*-*trnC* | CNS | 26,942 | A | – | – |
| 11 | SNP | LSC/*trnC*-*petN* | CNS | 27,715 | G | T | G |
| 12 | SNP | LSC/*trnE*-*trnT* | CNS | 31,611 | A | T | A |
| 13 | SNP | LSC/*trnT*-*psbD* | CNS | 32,314 | A | T | A |
| 14 | SNP | LSC/*trnT*-*psbD* | CNS | 32,709 | T | A | A |
| 15 | SNP | LSC/*trnT*-*psbD* | CNS | 33,453 | C | A | C |
| 16 | InDel | LSC/*trnG*-*trnfM* | CNS | 37,465–37,466 | – | – | CCCCAAGAAAAAAAGG TAATTAATTATTCTTT |
| 17 | InDel | LSC/*ycf3*-*trnS* | CNS | 45,458 | T | – | T |
| 18 | InDel | LSC/*ycf3*-*trnS* | CNS | 46,152–46,153 | – | – | T |
| 19 | SNP | LSC/*trnT*-*trnL* | CNS | 47,869 | C | A | C |
| 20 | SNP | LSC/*trnT*-*trnL* | CNS | 48,096 | A | C | A |
| 21 | SNP | LSC/*trnT*-*trnL* | CNS | 48,097 | G | T | T |
| 22 | SNP | LSC/*trnL*-*trnF* | CNS | 49,580 | T | C | T |
| 23 | SNP | LSC/*trnF*-*ndhJ* | CNS | 50,344–50,348 | AAAAG | AAAAG | CTTTT |
| 24 | SNP | LSC/*trnV*-intron | CNS | 53,230 | C | A | C |
| 25 | InDel | LSC/*accD*-*psaI* | CNS | 59,997–60,008 | AATTTATTTTTA | – | AATTTATTTTTA |
| 26 | InDel | LSC/*psaI*-*ycf4* | CNS | 60,623 | T | – | – |
| 27 | InDel | LSC/*petA*-*psbJ* | CNS | 64,224 | G | – | G |
| 28 | InDel | LSC/ *petA*-*psbJ* | CNS | 64,355–64,356 | – | T | – |
| 29 | InDel | LSC/*psbE*-*petL* | CNS | 66,485 | A | – | – |
| 30 | SNP | LSC/*trnP*-*psaJ* | CNS | 67,723 | C | C | T |
| 31 | InDel | LSC/*trnP*-*psaJ* | CNS | 67,834–67,835 | – | TAGTAA | – |
| 32 | SNP | LSC/*psaJ*-*rpl33* | CNS | 68,408 | A | A | T |
| 33 | InDel | LSC/*rps18*-*rpl20* | CNS | 69,490 | A | – | A |
| 34 | InDel | LSC/ *rps18*-*rpl20* | CNS | 69,491 | A | – | – |
| 35 | SNP | LSC/*rpl20*-*rps12* | CNS | 70,254 | A | G | G |
| 36 | SNP | LSC/*rpl20*-*rps12* | CNS | 70,519 | A | A | T |
| 37 | InDel | LSC/*rps12*-*clpP* | CNS | 70,966–70,967 | – | T | – |
| 38 | SNP | LSC/*rps12*-*clpP* | CNS | 70,999 | G | G | T |
| 39 | InDel | LSC/*clpP*-intron | CNS | 71,668–71,669 | – | T | – |
| 40 | SNP | LSC/*clpP*-intron | CNS | 71,681 | C | A | C |

| Number | Type | Position | Location | Nucleotide position[b] | F. × ananassa 'Benihoppe' | F. chiloensis (GP33) | F. virginiana (O477) |
|--------|------|----------|----------|------------------------|---------------------------|----------------------|----------------------|
| 41 | SNP | LSC/clpP-intron | CNS | 72,808 | T | C | T |
| 42 | InDel | LSC/psbT-psbN | CNS | 75,456–75,457 | – | CATTATCTC AATTGAAAGT | – |
| 43 | SNP | LSC/petD-intron | CNS | 78,077 | G | A | G |
| 44 | SNP | LSC/rpl36-rps8 | CNS | 80,856 | C | G | C |
| 45 | InDel | LSC/rpl14-rpl16 | CNS | 82,300 | T | – | – |
| 46 | SNP | LSC/rpl16-intron | CNS | 82,928 | G | T | T |
| 47 | SNP | LSC/rpl16-intron | CNS | 83,676 | T | T | C |
| 48 | SNP | IR/rps12-trnV | CNS | 100,249 | C | A | C |
| 49 | SNP | IR/rrn5-trnR | CNS | 109,248 | G | T | G |
| 50 | SNP | IR/trnN-ycf1 (short) | CNS | 110,162 | A | G | A |
| 51 | SNP | SSC/ndhF-rpl32 | CNS | 113,838 | T | A | T |
| 52 | SNP | SSC/rpl32-trnL | CNS | 114,675 | T | T | A |
| 53 | SNP | SSC/ndhD-psaC | CNS | 118,166 | C | A | C |
| 54 | InDel | SSC/psaC-ndhE | CNS | 118,599–118,600 | – | A | – |
| 55 | SNP | SSC/ndhA-intron | CNS | 122,406 | C | A | C |
| 56 | SNP | LSC/accD | Gene | 58,891 | C | T | C |
| 57 | SNP | SSC/ndhF | Gene | 113,349 | A | G | A |
| 58 | SNP | SSC/ndhH | Gene | 123,504 | T | C | T |
| 59 | SNP | LSC/petB | Gene | 77,457 | G | G | T |
| 60 | SNP | LSC/psbA | Gene | 676 | A | A | G |
| 61 | SNP | LSC/rpoB | Gene | 25,334 | T | G | G |
| 62 | SNP | LSC/rps8 | Gene | 81,522 | T | T | C |
| 63 | SNP | SSC/ycf1 | Gene | 125,275 | G | T | G |
| 64 | SNP | SSC/ycf1 | Gene | 128,610 | G | C | G |
| 65 | SNP | SSC/ycf1 | Gene | 129,102 | G | G | T |
| 66 | SNP | SSC/ycf1 | Gene | 129,303 | C | A | C |
| 67 | SNP | LSC/ycf4 | Gene | 61,151 | G | A | A |

**Notes.**
[a]CNS, Non-coding sequences which containing intergenic spacer region and introns.
[b]Nucleotide position is referenced to the chloroplast genome of F. × ananassa 'Benihoppe'.

*F. virginiana* and different from *F. chiloensis*. Our results are in accordance with previous studies (*Salamone et al., 2013*; *Honjo et al., 2009*) and indicate that the sequence identity between *F. × ananassa* 'Benihoppe' and *F. virginiana* (O477) at the chloroplast level is higher than between *F. × ananassa* 'Benihoppe' and *F. chiloensis* (GP33).

## cpDNA markers and sequence polymorphisms in *Fragaria*

Non-coding regions (introns and intergenic spacers), harboring more sequence divergence, are not subject to the functional constraints that could extend the utility of a molecule at lower taxonomic levels (*Small et al., 1998*; *Shaw et al., 2007*). At least six non-coding regions of cpDNA were previously examined in phylogenetic and ancestry studies of *Fragaria*. Universal primers targeting the *trnL-trnF* region (*Taberlet et al., 1991*) were

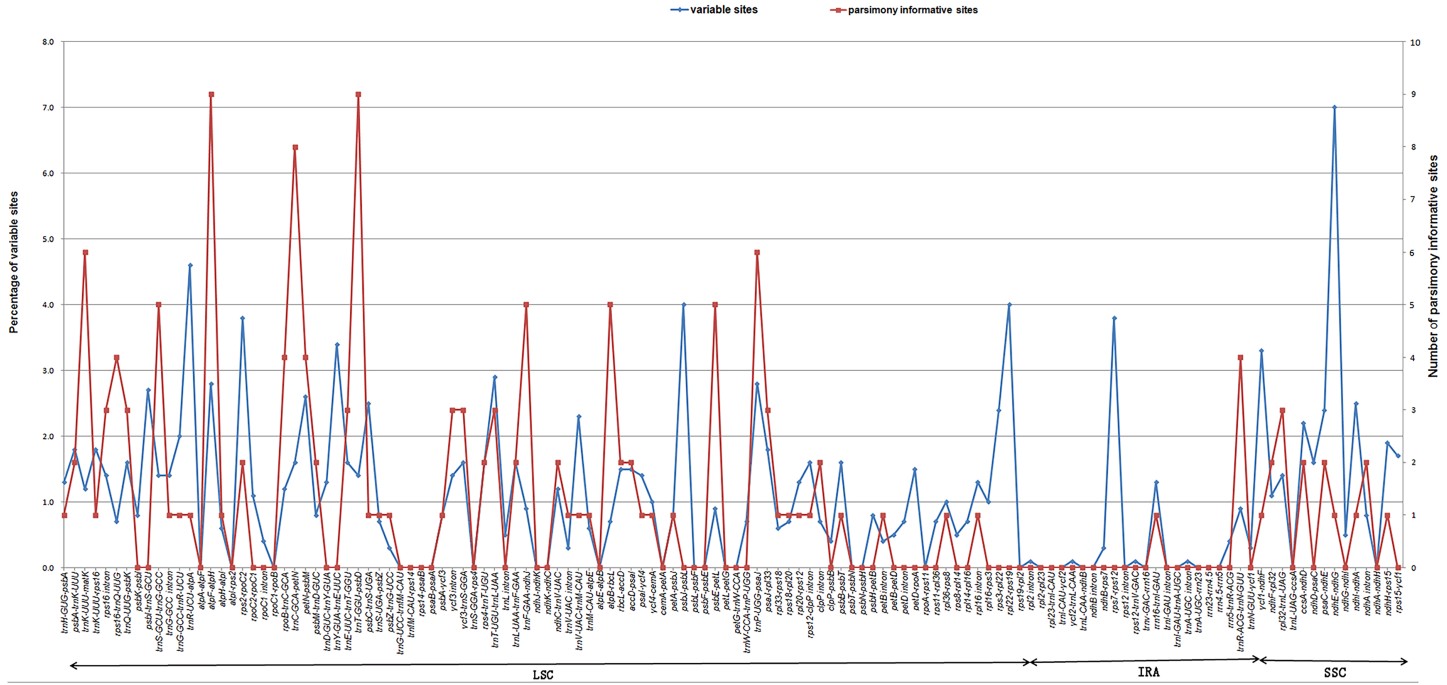

**Figure 5** Percentage of variable sites and number of parsimony-informative sites in non-coding regions across the ten *Fragaria* chloroplast genomes.

used in previous studies to investigate 14 species of *Fragaria* (*Potter, Luby & Harrison, 2000*) and 8 diploid species from this genus (*Sargent, 2005*). The *trnT-trnL*, *atpB-rbcL*, *psbA-trnH*, *psbJ-psbF* and *rps18-rpl20* regions have also been employed to detect sequence polymorphisms in *Fragaria* (*Sargent, 2005*; *Lin & Davis, 2000*). However, due to the small group of taxa sampled or a low level of sequence variation in these regions, the phylogenetic resolution within *Fragaria* has been limited (*Rousseau-Gueutin et al., 2009*).

To examine which regions might be applied for *Fragaria* phylogenetic analysis, all of the non-coding regions among ten *Fragaria* chloroplast genomes were aligned, and sequence divergence was calculated (Fig. 5). The results showed that *ndhE-ndhG* exhibits the highest rate of variation (7%), while *atpF-atpH* and *trnT-psbD* exhibit the most PIS. However, only the 6 intergenic regions (*trnK-matK*, *trnS-trnG*, *atpF-atpH*, *trnC-petN*, *trnT-psbD*, and *trnP-psaJ*) display a percentage of variable sites higher than 1% and more than five PIS (Fig. 5), indicating the low variation of chloroplast genomes in *Fragaria*. Interestingly, these intergenic regions are all located in the LSC region, whose sequence has been noted to be less conserved in those of IR and SSC regions and has consequently been used for phylogenetic analysis at low taxonomic levels (*Njuguna, 2010*).

To examine the phylogenetic applications of the six fast-evolving DNA regions, an MP tree was constructed for each molecular marker from the ten *Fragaria* species (File S6). The results revealed that none of each region was efficient in resolving the relationships among the examined samples. However, the combined regions strongly supported *F. iinumae* (fc199s5) and *F. pentaphylla* (KY434061) had the closest phylogenetic relationship.

Furthermore, our results showed *F. vesca* ssp. *vesca* (Hawaii 4) was closer to *F. vesca* ssp. *bracteata* (LNF40), which was similar to *Govindarajulu et al. (2015)*. The STEMhy and PhyloNet results showed a greater contribution of *F. iinumae* than *F. vesca* to the ancestry of the octoploids (*Kamneva et al., 2017*). *Vining et al. (2017)* used POLIMAPS to resolve *F. × ananassa* chromosomal regions derived from diploid ancestor *F. vesca*. Our results couldn't infer which one was the ancestor of *F. × ananassa* 'Benihoppe' (KY358226) at present. Further studies with a broad sampling scheme need to be conducted to test the efficiency of these six identified regions in phylogenetic analysis of *Fragaria*.

## CONCLUSIONS

This study provides the first report of the complete chloroplast genome sequence of *F. × ananassa* 'Benihoppe'. Comparison with nine Rosaceae species revealed higher sequence variation in SC regions compared with IR regions in both coding and non-coding regions, and the gene order, gene content and genome structure were found to be similar to those of other sequenced *Fragaria* species, especially *F. virginiana* (O477) and *F. chiloensis* (GP33), demonstrating low variation among *Fragaria* chloroplast genomes. However, IR contraction is observed in *F. × ananassa* 'Benihoppe', and several SNPs and InDels identified among three octoploid strawberries can be utilized for diversity analyses. Six non-coding regions (*trnK-matK*, *trnS-trnG*, *atpF-atpH*, *trnC-petN*, *trnT-psbD* and *trnP-psaJ*) may be useful for phylogenetic analysis of the genus *Fragaria*. The chloroplast genome of *F. × ananassa* 'Benihoppe' may also provide important information for research related to the chloroplast transgenic engineering of cultivated strawberry.

### Funding
The research was supported by the Jiangsu Province Agriculture Science and Technology Innovation Fund (no. CX (15) 1029). The funders had no role in study design, data collection and analysis, decision to publish, or preparation of the manuscript.

### Grant Disclosures
The following grant information was disclosed by the authors:
Jiangsu Province Agriculture Science and Technology Innovation Fund: CX (15) 1029.

### Competing Interests
The authors declare there are no competing interests.

### Author Contributions
- Hui Cheng performed the experiments, analyzed the data, wrote the paper, prepared figures and/or tables.
- Jinfeng Li, Binhua Cai and Zhihong Gao contributed reagents/materials/analysis tools.
- Hong Zhang performed the experiments.

- Yushan Qiao conceived and designed the experiments, analyzed the data, reviewed drafts of the paper.
- Lin Mi conceived and designed the experiments.

## DNA Deposition

The following information was supplied regarding the deposition of DNA sequences:
GenBank accession number: KY358226.

## Data Availability

The raw data can be found at GenBank: JN884816, JN884817, JF345175, KC507755, KY434061, KX768420, KU851961, AP012207, HQ336405.

## Supplemental Information

Supplemental information for this article can be found online at http://dx.doi.org/10.7717/peerj.3919#supplemental-information.

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
