# Peer review of "The complete chloroplast genome sequence of strawberry (Fragaria × ananassa Duch.) and comparison with related species of Rosaceae"

_PeerJ, doi:10.7717/peerj.3919_

## Round 0.1 · original submission · Minor Revisions

Please review the comments of the two reviews below. I think they have done a good job and provide some good suggestions.

·

Basic reporting

Sufficient field background information was provided. In general, writing was clear.

Experimental design

This work will provide valuable information through the complete sequence of the chloroplast of the cultivated octoploid. This will allow to improve chloroplast genetic engineering and to discard this genome in whole genome sequencing done on nuclear DNA or RNA. The research question was clear, relevant and performed accordingly. The applied methods were well described and detailed.

Validity of the findings

Data was well analyzed, statistically sound. Conclusion were well stated, and linked to original research question.

Additional comments

The cultivated strawberry (Fragaria x ananassa Duch.) is one of the youngest domesticated plants, developed from chance hybridization between two western hemisphere octoploid species (F. chiloensis and F. virginiana), which becomes one of an important fruit crop in the world to date. However, little is known about the chloroplast genome of the species that gave rise to this important fruit crop.
In the manuscript, the authors present the complete sequence of the chloroplast genome of the cultivated strawberry. For generating the complete genome, they performed a combination of de novo assembly and reference-guided mapping of contigs. The complete genome sequence provides a new reference for the evolution and genetic diversity of plants.
Then, they compared the chloroplast genome structure and organisation of F. × ananassa cv. Benihoppe with ones of other Rosaceae species. Finally, they performed a cpDNA markers and sequence polymorphisms analysis using data from ten species belonging to Fragaria genus.
This work will provide valuable information through the complete sequence of the chloroplast of the cultivated octoploid. This will allow to improve chloroplast genetic engineering and to discard this genome in whole genome sequencing done on nuclear DNA or RNA. The research question was clear, relevant and performed accordingly. The applied methods were well described and detailed. The article is in the scope of the journal Peer J. Minor version is required to considered the following comments.
Compared with the nuclear genome, the chloroplast genome is small, and the rate of nucleotide substitutions is so low that the chloroplast genome is considered to be an ideal system for studies on phylogeny and population genetics (Wei et al., 2005). I do not think low rate of substitution is good for population study, definitely is good for phylogenetic analysis.

There are 130 genes in the chloroplast genome of F. × ananassa cv. Benihoppe, 112 of which are unique. What do you mean by the “unique”?

Six fast-evolving DNA regions were individually and combined to construct phylogeny of ten Fragaria species. More discussion on the phylogenetic analysis with previous ones is suggested to be expended.

Please explore the basis for disagreement between some of their own findings and a highly relevant prior study involving Benihoppe.

Reviewer 2 ·

Basic reporting

An additional chloroplast assembly exists that is not mentioned which provides results of a partial 130 kb chloroplast assembly of F. vesca as submitted GenBank GU363535.1 and described in Davis TM, Shields ME, Reinhard AE, Reavey PA, Lin J, Zhang H, Mahoney LL, Bassil NV, Martin R. 2010. Chloroplast DNA inheritance, ancestry, and sequencing in Fragaria. Acta Horticulturae 859: 221-228.

Line 13 Fragaria chloroplast exhibits low variation on what basis? What is the evidence? Or is this based on work of Wei and al 2005 for chloroplast genomes in general or based on Zhang and al 2017 for Rosasceae? The statement is not clear to this reviewer.
Line 24 What did the comparative analysis consist of briefly?
Line 29. How were F. chiloensis and F.virginiana compared to cv Benihoppe and which F. chiloensis and F. virginiana? For ease, the PI# should be used to indicate each of the accessions used in the study for Fragaria and for all of the Rosaceae species.
Introduction
Line 57 Hirtula rose as distinct from rose?
Line 58 Assembly of five Fragaria species (Shulaev et al., 2011; Hirakawa et al., 2014). There should be more such as Tennessen et al including octoploid and F. vesca ssp bracteata publications.
Line 60 Assembly of nuclear genomes are still a work in progress and not finished. Maybe state that drafts assemblies have been released?
Line 78-79 Provide references to support that Fragaria has limited variation in chloroplast sequences.
Line 86 cp genomes of F. vesca by TMD. The following reference also provides results of a partial 130 kb chloroplast assembly of F. vesca and reference to a GenBank submission. Davis TM, Shields ME, Reinhard AE, Reavey PA, Lin J, Zhang H, Mahoney LL, Bassil NV, Martin R. 2010. Chloroplast DNA inheritance, ancestry, and sequencing in Fragaria. Acta Horticulturae 859: 221-228.
Line 98 Transgenic Benihoppe nuclear genome released versus reported?
Line 239 - 244. Sampling of 8x and 2x species is too limited to make broad statements on the species, statement and comparisons should be made to those specific accessions that were compared.
Line 334 comparison to how many F. virginiana and F. chiloensis and which ones?

Experimental design

Inclusion of additional accessions of especially F. virginiana and of F. chiloensis to represent the diversity of the species would permit the kind of conclusions drawn in the study. As only one cultivar was sequenced, the ancestry of the cultivar might be useful as insight into the possible similarity to cultivars in use other countries.

Validity of the findings

Any plant accessions used in the study should have Plant Introduction numbers and if Plant Introduction numbers do not already exist, the authors should submit them to a germplasm repository and obtain plant introduction numbers. There are subspecies of both F. virginiana and F. chiloensis and variation within the subspecies, so the specifics of what material was used is critical. There is also variation in F. vesca and F. iinumae and so conclusions should be limited to the comparison made with the specific accessions used in this study and not be drawn to encompass the entire species.

Additional comments

If the above comments are addressed, I recommend the article for publication.

---

## Round 0.2 · accepted · Accept

Thank you for making the reviewers changes, I believe it helped to make you paper better.